# The Prognostic Value of the Fibrinogen-Albumin-Ratio Index (FARI) in Patients with Advanced Vulvar Cancer

**DOI:** 10.3390/jpm12111882

**Published:** 2022-11-10

**Authors:** Arina Onoprienko, Gerda Hofstetter, Tim Dorittke, Christine Bekos, Christoph Grimm, Mariella Polterauer, Thomas Bartl, Stephan Polterauer

**Affiliations:** 1Department of Obstetrics and Gynecology, Division of General Gynecology and Gynecologic Oncology, Medical University of Vienna, 1090 Vienna, Austria; 2Department of Pathology, Medical University of Vienna, 1090 Vienna, Austria; 3Karl Landsteiner Institute for General Gynecology and Experimental Gynecologic Oncology, 1090 Vienna, Austria

**Keywords:** vulvar cancer, biomarker, albumin, fibrinogen, prognosis

## Abstract

The present study aims to evaluate the pretherapeutic Fibrinogen-Albumin-Ratio Index (FARI), as currently reliable biomarkers to predict therapy response and prognosis of patients with advanced vulvar cancer are missing. Data of 124 consecutive patients, who underwent primary resection for vulvar cancer ≥ pT1b, were retrospectively analyzed. Associations between the FARI and disease recurrence were assessed fitting receiver operating characteristics (ROC) and binary logistic regression models; univariate and multivariable Cox regression models for disease-specific survival (DSS) and progression-free survival (PFS) were performed. A pretherapeutic low FARI cut at its median (<9.67) is significantly associated with younger age (65.5 vs. 74.0 years) and higher risk of recurrence (52.4% vs. 26.2%). The ROC analysis calculates the area under the curve (AUC) of the FARI for a PFS < 6 months of 0.700 and for a DSS < 12 months of 0.706, outperforming fibrinogen and albumin alone. The FARI remained independently predictive for PFS (HR 0.84, 95% CI [0.99–1.03], *p* = 0.009) and DSS (HR 0.82, 95% CI [0.70–0.99], *p* = 0.019), also in multivariable survival analysis. Despite the FARI’s promising predictive and prognostic value, however, further elucidation of its precise mode of action is warranted before clinical application as it appears to rely only on subtle changes of fibrinogen levels.

## 1. Introduction

Vulvar cancer accounts for 5% of gynecological malignancies with increasing incidence and mortality rates. Even though diagnosed at an average age of 68 years, a trend towards younger age at primary diagnosis and an increasing association with human papillomavirus (HPV) infection is being observed [1,2,3]. In Austria, the incidence of vulvar cancer is comparable with other industrial nations with 173 new diagnoses reported in 2020 [4]. Whereas early tumor stages can be effectively treated by local wide resection, more advanced stages are associated with poor oncologic outcome and often require more complex surgical procedures, frequently including adjuvant radiotherapy and/or chemotherapy. Evidence on optimal adjuvant or even recurrent therapy strategies, however, remains scarce, as the hitherto low incidence of this specific cancer type has been hampering clinical research until recently. To address these limitations and to foster more personalized therapeutic approaches, several clinical and pathological biomarkers to identify patients at risk of therapy-related adverse events or poor prognosis have been proposed in recent years [5,6,7,8]. Despite clinical tumor features as the number of positive inguinal lymph nodes, depth of invasion and FIGO stage remain the clinically most important prognostic factors of vulvar cancer, the individual outcome varies significantly depending on patient age and comorbidities [9,10].

Recently, the Fibrinogen-Albumin Ratio Index (FARI), the ratio of plasma fibrinogen and plasma albumin, was suggested as a readily available biomarker of systemic inflammation. The FARI demonstrated promising prognostic accuracy in several different cancer types, including colorectal cancer, cholangiocarcinoma, non-small cell lung cancer, and chronic lymphocytic leukemia [11,12,13,14]. High FARI values were also associated with unfavorable prognosis in ovarian cancer and cervical cancer patients [15,16]. To our knowledge, the prognostic value of the FARI has not been assessed for patients with vulvar cancer to date. The present study therefore aims to investigate both the predictive value for therapy response and the prognostic value for patient survival of the pre-treatment FARI in patients with advanced vulvar cancer.

## 2. Materials and Methods

The current study was designed as a single-center retrospective chart review analyzing data of all consecutive patients with invasive vulvar cancer with International Federation of Gynecology and Obstetrics (FIGO) stages IB–IVA who underwent primary surgical resection at the Department of General Gynecology and Gynecological Oncology of the Medical University of Vienna between 2000 and 2020 (*n* = 204).

All patients diagnosed with vulvar cancer based on histopathologic examination after primary surgery, and for whom follow-up data at our institution was available, were evaluated. Patients with early staged disease (pT1a according to final histopathology) and those with any kind of neoadjuvant therapy were not considered for inclusion; patients with rare histotypes (Morbus Paget, neuroendocrine, malignant melanoma), patients with any other active secondary malignancy within five years from diagnosis, patients for whom no preoperative hematologic test result including fibrinogen and albumin levels could be obtained, and patients who did not undergo oncologic follow-up at our department were excluded from final analysis (Figure 1).

According to our institution’s internal standards, every patient planned for surgery for vulvar cancer had to obtain a hematologic test result at maximum 14 days before the date of surgery, which was analyzed in the present study. Thereof, the FARI was calculated as the ratio of fibrinogen (g/L) divided by albumin (g/L) and multiplied by 100 [17]. The present study was conducted in compliance with the declaration of Helsinki and approved by the ethics committee of the Medical University of Vienna (IRB 1901/2017).

All vulvar cancer patients underwent surgical tumor resection, with or without lymph node assessment as previously described [18]. Sentinel lymph node assessment was performed using blue dye and starting from 2009, with or without technetium-99. Application of indocyanine green replaced former methods starting from 2017. In cases of lymph node involvement, systemic inguinal-femoral lymph node dissection was performed, and postoperative radiotherapy was conducted at the Department of Radiotherapy of the Medical University of Vienna if macrometastases were found. After primary cancer treatment, all patients were included into our department’s routine oncologic follow-up program, which comprises regular clinical examination and imaging on over 10 years.

Statistical analysis was performed using SPSS^®^ (IBM Corp. Released 2016. IBM SPSS Statistics for Windows, v24.0. IBM Corp., Armonk, NY, USA) for Windows and R 3.6.3 (R Core Team (2020). R: A language and environment for statistical computing. R: Foundation for Statistical Computing, Vienna, Austria. https://www.R-project.org, accessed on 15 June 2022). Patient data were analyzed by descriptive statistics. Categorical variables were described using percentages and medians with interquartile range. For all statistical tests, *p*-values below 0.05 were considered statistically significant. To identify associations between the FARI and clinicopathological variables, Student’s t-test, chi-squared test, or one-way analysis of variance were performed, where appropriate. To assess the predictive value of the FARI on disease recurrence as compared to fibrinogen and albumin levels alone, receiver operating characteristics were calculated. Results were validated by fitting binary logistic regression models with the endpoint of disease recurrence, where applicable. Survival analyses were performed for the endpoints of progression-free survival (PFS) and disease-specific survival (DSS) with a follow-up data cut-off of 31 December 2021. DSS and PFS were defined as the time between primary surgery to cancer recurrence as documented by our department’s internal interdisciplinary tumor board and vulvar cancer-related death, respectively. Univariate survival analysis was performed using log ranks-test and patient’s survival was visualized by Kaplan–Meier curves. A multivariate Cox proportional hazard model was established with a stepwise hierarchical selection approach at a significance level of *p* < 0.05. Hazard ratios were indicated with a two-sided confidence interval of 95%, where appropriate. Survivors were censored at the last timepoint of available follow-up data.

## 3. Results

### 3.1. Descriptive Characteristics

In total, 124 patients were included into final statistical analysis. Pretherapeutic patient baseline patients’ characteristics are given in Table 1. A FARI Cut at the median (</≥9.67), a lower FARI was significantly associated with younger patient age (65.5 vs. 74.0 years, *p* = 0.016) and a higher risk of recurrence (52.4% vs. 26.2%, *p* = 0.003).

Of note, both the FARI (r = 0.297, *p* = 0.001) and fibrinogen (r = 0.328, *p* = 0.001), but not albumin (r = −0.128, *p* = 0.156) demonstrated a significant correlation with patient age. In the overall patient cohort, median PFS was 15.0 (5.0–54.5) months, median overall survival was 26.0 (9.0–86.0) months.

### 3.2. The FARI Predicts Early Disease Recurrence and Worse Disease-Specific Survival

The receiver operating characteristic (ROC) analysis calculated the area under the curve (AUC) of the FARI for disease recurrence of 0.605 (0.505–0.706), for a PFS < 6 months of 0.700 (0.596–0.804) and for a DSS < 12 months of 0.706 (0.602–8.11) (Figure 2a,b) to an AUC of 0.663 (0.555–0.770) (PFS < 6 months), and 0.625 (0.573–0.791) (DSS < 12 months) for fibrinogen and 0.663 (0.524–0.724) (PFS < 6 months) and 0.520 (0.340–0.565) (DSS < 12 months) for albumin.

In a univariate binary logistic regression model applying continuous variables with the endpoint of disease recurrence, neither the FARI (OR 0.88, 95% CI [0.76–1.02], *p* = 0.090), nor albumin (OR 1.05, 95% CI [0.97–1.14], *p* = 0.230), nor fibrinogen (OR 1.00, 95% CI [1.00–1.00], *p* = 0.265) demonstrated significant results. Fitting univariate binary logistic regression models applying continuous variables with the endpoints of a PFS < 6 months for early disease recurrence and a DSS < 12 months for worse survival, the FARI was associated with both a PFS < 6months (OR 1.37, 95% CI [1.12–1.58], *p* = 0.002) and a DSS < 12 months (OR 1.35, 95% CI [1.10–1.64], *p* = 0.004). In line, fibrinogen was associated with a PFS < 6 months (OR 1.01, 95% CI [1.00–1.01], *p* = 0.012) and a DSS < 12 months (OR 1.00, 95% CI [1.00–1.01], *p* = 0.006). Albumin was associated with a PFS < 6 months (OR 0.90, 95% CI [0.82–0.99], *p* = 0.04), but not with a DSS < 12 months (OR 0.95, 95% CI [0.86–1.03], *p* = 0.183).

### 3.3. The FARI Is Associated with Progression-Free and Disease-Specific Survival

The FARI was associated with both PFS (HR 0.84, 95% CI [Cl 0.74–0.96], *p* = 0.011) and DSS (HR 0.85, 95% CI [0.72–1.00], *p* = 0.049) in univariable COX- regression. Results could be confirmed in multivariable analyses for PFS (HR 0.84, 95% CI [Cl 0.99–1.03], *p* = 0.009) as compared to FIGO-staging and age at therapy, and DSS (HR 0.82, 95% CI [0.70–0.99], *p* = 0.019) as compared to FIGO-staging, CRP (mg/dl) and age at therapy. Of note, CRP (mg/dl) was independently associated with DSS. Preoperative albumin did not demonstrate significant associations with PFS or DSS; in contrast, preoperative fibrinogen was associated with DSS, but not PFS (Table 2). Kapan–Meier curves including confidence intervals depicting a significantly increased PFS (*p* < 0.0001) and DSS (*p* = 0.0023) in the subgroups with a preoperatively increased FARI cut at its median are given in Figure 3a,b.

## 4. Discussion

A decreased FARI was independently associated with impaired PFS and DSS in vulvar cancer patients undergoing upfront surgery. The FARI thereby appears to outperform the prognostic value of both isolated pretreatment albumin as well as fibrinogen measurements. A decreased FARI was more likely to be present in younger patients who experienced earlier disease recurrences.

The present study is the first to evaluate the FARI in vulvar cancer patients; results, however, appear to contrast the previously published literature for other gynecologic cancers. Whereas present data suggest a decreased FARI, i.e., decreased fibrinogen and elevated albumin, to be associated with impaired survival, previous studies of ovarian, cervical, and breast cancer report either an elevated FARI or a decreased albumin-to-fibrinogen ratio, i.e., elevated fibrinogen and decreased albumin, to be associated with impaired survival.

For patients with ovarian cancer, Li et al. developed a scoring system for fibrinogen and albumin aberrations and described elevated fibrinogen and decreased albumin levels to be associated with poor prognosis in patients undergoing primary cytoreductive surgery [19]. In line, Yu et al. reports that a decreased albumin-to-fibrinogen ratio is associated with both poor PFS and OS in patients undergoing intervention debulking surgery after neoadjuvant chemotherapy [15]. For cervical cancer patients, a recent study reports that an elevated FARI is significantly associated with impaired PFS and OS in patients undergoing radical hysterectomy followed by adjuvant chemotherapy [16]. Another study indicates that a low albumin-to-fibrinogen ratio to be associated with higher tumor size and stage [20]. For breast cancer patients, a study reports an elevated FARI to be independently associated with impaired OS with the strongest effect in luminal A-like subtypes. The predictive value of the FARI outperformed either fibrinogen and albumin levels alone [21]. For vulvar cancer patients, no evidence on the prognostic value of the FARI is available to date.

With regard to either parameter alone, preoperative hypoalbuminemia < 3.5 g/dL was previously associated with both postoperative complications and impaired survival in patients with vulvar cancer [22,23]. Low albumin levels could therefore be a consequence of systematic inflammatory response and malnutrition in advanced stage tumors; in line, albumin could be established as both predictive for therapy response and prognostic for survival in several gynecologic cancers [8,24,25]. Fibrinogen was previously reported to promote cancer cell growth and progression; fibrinogen levels were observed to be associated with tumor size and tumor invasion [7]. However, no independent data on the prognostic value of fibrinogen in vulvar cancer patients is available to date. In light of present observations, it is to be noted that differences in albumin levels between the cohorts (FARI < 9.67: median albumin 42.9 g/L; FARI ≥ 9.67: median albumin 40.3 g/L) are not to be considered clinically relevant and are beyond established cut-offs of hypoalbuminemia. It may therefore be that the observed prognostic value of the FARI appears to rely on changes of fibrinogen levels. Fibrinogen was previously associated with increased age and high tumor burden as frequently observed in advanced ovarian cancer and cervical cancer patients [26,27]. Patient age, however, did not demonstrate prognostic value in multivariate analysis. Moreover, vulvar cancer patients are often diagnosed in a localized stage; as no primarily metastasized patient was included in the group with a FARI ≥ 9.67, it cannot be ruled out that elevated fibrinogen levels are to be explained by other, yet unidentified confounders.

Even though the definition of readily available independent prognostic factors such as the FARI is of high clinical interest to improve more personalized therapeutic strategies, further research is necessary before the implementation of the FARI could be in clinical routine. As the mode of action appears to rely on rather subtle changes of fibrinogen levels, its clinical application is not to be recommended outside clinical studies until its mode of action is fully elucidated.

The present study faces a number of limitations. As typical for retrospective analyses, the lack of random patient assignment, patient selection, and potentially incomplete data acquisition alleviate its clinical applicability. Treatment standards have changed within the study period and due to the low incidence of vulvar cancer that can be regarded an orphan disease, numbers in both of our groups are limited. Moreover, the FARI could only be assessed before surgery, whether or not its postoperative course may improve its prognostic value remains unclear. Due to the low incidence of this specific disease, the cohort includes patients starting from the year 2000 to achieve adequate statistical power. Therefore, no complete data on the HPV-status of the respective tumors is available. To address this clinically highly relevant question, we re-reviewed all histopathologic results for information regarding HPV-related alterations (e.g., description of koilocytes). Due to obvious methodological flaws, this observation is, however, to be interpreted critically. As HPV-driven tumors are hypothesized to demonstrate higher immunogenicity, more detailed data relating the FARI-value and respective HPV-status would have been of interest. Cohorts are slightly skewed, showing a higher FARI to be associated with older age and morbidity. It remains unclear whether comorbidity is causally linked to the FARI or if a higher percentage of morbid patients were reported due to older age (while the association of advanced age and higher fibrinogen is well-established). As neither ECOG or patient age was significantly associated with PFS or DSS in survival analysis, we consider the confounding effect of age and/or comorbidity for this research question limited. Contrarily, as the FARI appears to be resilient against age and/or comorbidity in survival analysis, it could prove clinically valuable, particularly in older or morbid patients. However, this is the first report on the use of the FARI in patients with vulvar cancer and our findings may generate interesting hypothesis for further validation.

## 5. Conclusions

Despite that the FARI appears to demonstrate a promising prognostic value in patients with advanced vulvar cancer, its unclear mode of action limits its clinical applicability. Further clinical studies are necessary before clinical implementation can be recommended.

## Figures and Tables

**Figure 1 jpm-12-01882-f001:**
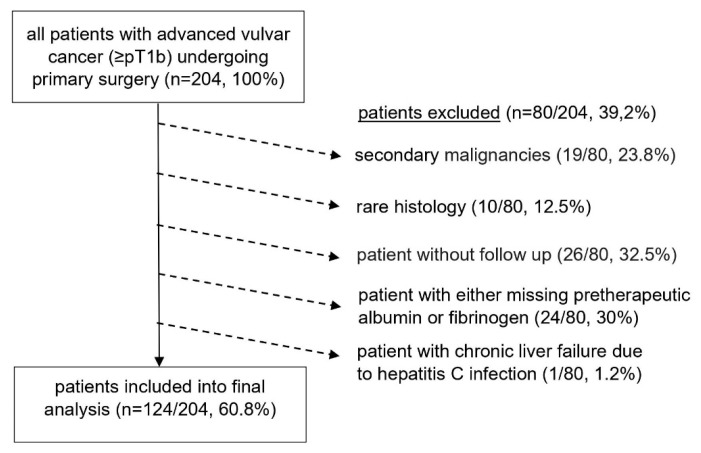
Flowchart depicting the constitution of the patient cohort evaluated at final analysis (*n* = 124). Out of 204 patients with advanced vulvar cancer (≥pT1b) who underwent primary surgery initially evaluated, 80 were excluded for not meeting primary inclusion criteria: due to secondary malignancy (19), rare histotype of tumor (10), chronic liver failure due to hepatitis C infection (1), missing follow-up (26) and either missing pretherapeutic albumin or fibrinogen (24).

**Figure 2 jpm-12-01882-f002:**
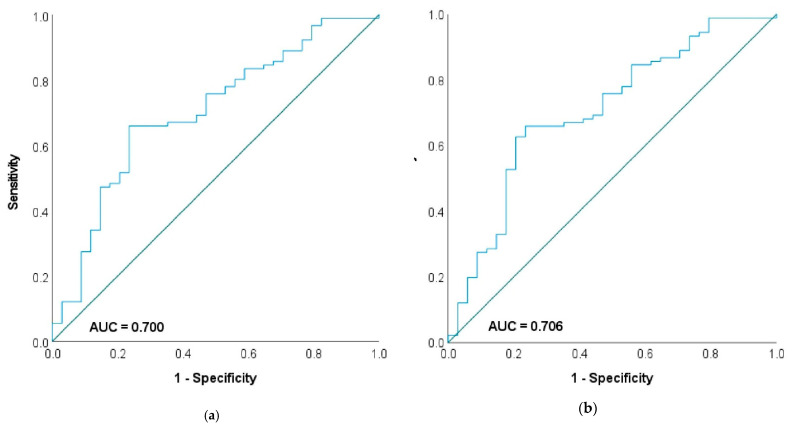
Receiver operating characteristic (ROC) analysis depicting the predictive value of the Fibrinogen-Albumin-Ratio (FARI) for a progression free survival (PFS) < 6 months (**a**) and disease-specific survival (DSS) < 12 months (**b**).

**Figure 3 jpm-12-01882-f003:**
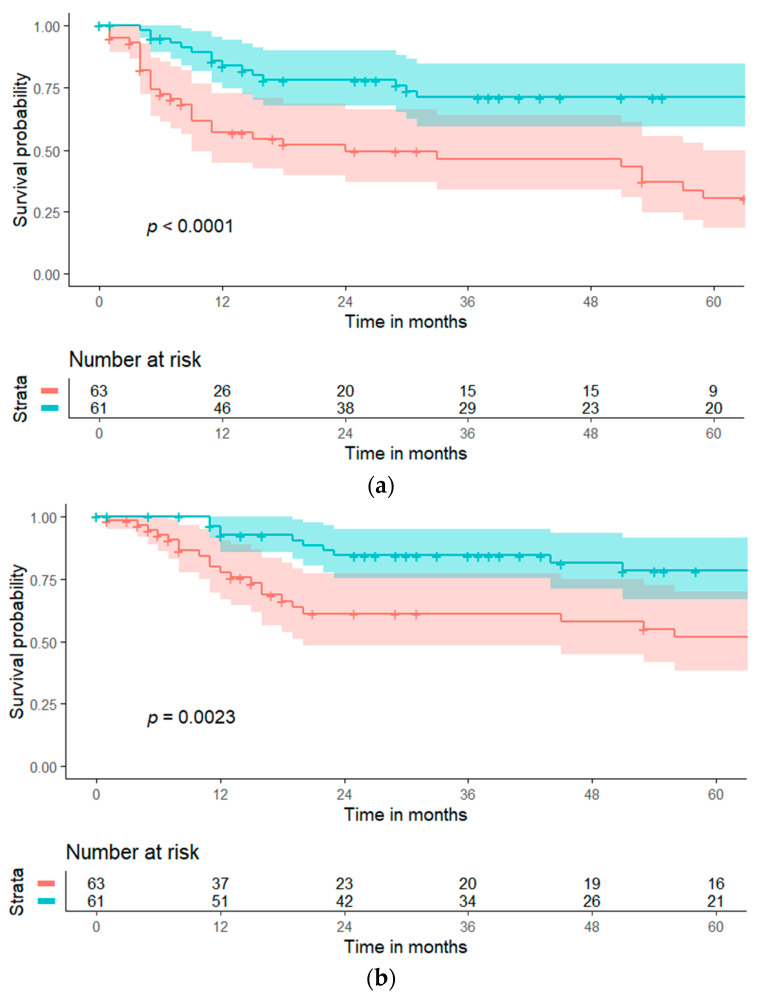
(**a**) Kaplan–Meier curve depicting progression-free survival (PFS) with confidence interval estimates at the timepoint of primary surgery distributed by a pre-therapeutic FARI. A high pre-therapeutic FARI (≥9.67) was associated with significantly longer PFS. The blue line depicts the cohort with a pretherapeutic FARI ≥ 9.67 (*n* = 61), the red line depicts the cohort with a pretherapeutic FARI < 9.67 (*n* = 63). (**b**) Kaplan–Meier curve depicting disease-specific survival (DSS) with confidence interval estimates at the timepoint of primary surgery distributed by a pre-therapeutic FARI. A high pre-therapeutic FARI (≥9.67) was associated with significantly improved DSS. The blue line depicts the cohort with a pretherapeutic FARI ≥ 9.67 (*n* = 61), the red line depicts the cohort with a pretherapeutic FARI < 9.67 (*n* = 63).

**Table 1 jpm-12-01882-t001:** Descriptive statistics of patients with advanced vulvar cancer undergoing primary surgery.

Variables	Overall Cohort	FARI < 9.67	FARI ≥ 9.67	*p*-Value
Number	124 (100%)	63 (50.8%)	61 (49.2%)	
Age at therapy (years)	69.0 (58.5–79.0)	65.5 (56.0–75.8)	74.0 (61.5–82.0)	0.016 *
ECOG				0.012 ^‡^
0	58 (46.8%)	34 (53.9%)	24 (39.3%)	
1	34 (27.4%)	18 (28.6%)	16 (26.2%)	
2	8 (6.5%)	1 (1.6%)	7 (11.4%)	
3	2 (1.6%)	0 (0%)	2 (3.3%)	
unknown	22 (17.7%)	10 (15.9%)	12 (19.8%)	
BMI	25.9 (22.5–31.2)	26.5 (22.7–31.3)	24.5 (21.7–30.7)	0.571 *
Histotype				0.207 †
Adenocarcinoma	7 (5.6%)	2 (3.2%)	5 (8.2%)	
Squamous cell carcinoma	117 (94.4%)	61 (96.8%)	56 (91.8%)	
HPV-associated tumor				0.789 †
yes	25 (20.2%)	12 (19%)	13 (21.3%)	
no	30 (24.2%)	16 (25.4%)	14 (23.0%)	
data not available	69 (55.6%)	35 (55.6%)	34 (55.7%)	
FIGO-stage				0.462 ^‡^
I	73 (58.9%)	37 (58.7%)	36 (59.0%)	
II	15 (12.1%)	6 (9.5%)	9 (14.8%)	
III	34 (27.4%)	18 (28.6%)	16 (26.2%)	
IV	2 (1.6%)	2 (3.2%)	0 (0%)	
recurrence				0.003 †
yes	49 (39.5%)	33 (52.4%)	16 (26.2%)	
no	75 (60.5%)	30 (47.6%)	45 (73.8%)	
Status at last FU				0.113 ^‡^
no evidence of disease	74 (59.7%)	33 (52.4%)	41 (67.2%)	
stable disease	3 (2.4%)	2 (3.2%)	1 (1.6%)	
progression	9 (7.2 %)	6 (9.5%)	3 (4.9%)	
intercurrent death	13(10.5%)	7 (11.1%)	6 (9.9%)	
cancer-related death	25 (20.2%)	15 (23.8%)	10 (16.4%)	
CRP (mg/dL)	0.42 (0.09–1.10)	0.27 (0.08–0.56)	0.79 (0.11–1.74)	0.067 *
fibrinogen (g/L)	4.0 (3.3–4.6)	3.4 (3.0–3.8)	4.6 (4.2–5.1)	<0.001 *
albumin (g/L)	38.8 (41.8–44.1)	42.9 (40.1–45.0)	40.3 (37.4–43.0)	<0.001 *

BMI, body mass index; CRP, C-reactive protein (mg/dl); ECOG, Eastern Cooperative Oncology Group; FIGO, International Federation of Gynecology and Obstetrics; FU, follow-up; * independent samples t-test; † chi-squared test; ‡ one-way ANOVA.

**Table 2 jpm-12-01882-t002:** Results of univariate and multivariable Cox regression model with respect to PFS on DSS. Multivariable models were fitted applying a stepwise hierarchical selection approach at a significance level *p* < 0.05. As younger age at therapy was associated with higher recurrence rates in descriptive statistics, the variable was included into multivariable analysis to rule out gross confounding.

	Progression-Free Survival (PFS)			
Parameters	Univariate Analysis		Multivariable Analysis	
	*p*-Value	HR (95% CI)	*p*-Value	HR (95% CI)
FARI	0.011	0.84 (0.74–0.96)	0.009	0.84 (0.99–1.03)
fibrinogen (g/L)	0.081	1.00 (0.99–1.00)	-	-
albumin (g/L)	0.088	1.06 (0.99–1.14)	-	-
CRP (mg/dL)	0.410	1.08 (0.90–1.28)	-	-
age at therapy (years)	0.371	1.01 (0.99–1.03)	0.422	1.01 (0.99–1.03)
FIGO I–II vs III–IV	<0.001	3.70 (2.11–6.50)	<0.001	3.47 (1.96–6.14)
ECOG 0–1 vs. 2–3	0.368	0.77 (0.43–1.37)	-	-
	**Disease-Specific Survival (DSS)**			
**Parameters**	**Univariate Analysis**		**Multivariable Analysis**	
	** *p* ** **-Value**	**HR (95% CI)**	** *p* ** **-Value**	**HR (95% CI)**
FARI	0.049	0.85 (0.72–1.00)	0.019	0.82 (0.70–0.99)
fibrinogen (g/L)	0.040	1.00 (0.99–1.00)	-	-
albumin (g/L)	0.842	1.01 (0.93–1.09)	-	-
CRP (mg/dL)	0.021	1.23 (1.03–1.46)	0.009	1.22 (1.05–1.42)
age at therapy (years)	0.410	1.01 (0.99–1.04)	0.255	1.01 (0.99–1.05)
FIGO I–II vs III–IV	<0.001	6.61 (3.11–14.06)	<0.001	6.58 (2.83–15.29)
ECOG 0–1 vs. 2–3	0.484	0.77 (0.38–1.59)	-	-

CRP, C-reactive protein (mg/dL); ECOG, Eastern Cooperative Oncology Group; FARI, Fibrinogen-Albumin Ratio Index; FIGO, International Federation of Gynecology and Obstetrics.

## Data Availability

The data presented in this study are available on request from the corresponding author.

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
