# Peer review of "The Prognostic Value of the Fibrinogen-Albumin-Ratio Index (FARI) in Patients with Advanced Vulvar Cancer"

_jpm, 2022, doi:10.3390/jpm12111882_

Round 1

Reviewer 1 Report

The manuscript “The Prognostic Value of the Fibrinogen-Albumin-Ratio Index (FARI) in Patients with Advanced Vulvar Cancer” by Arina Onoprienko  and co-authors to evaluate the pretherapeutic Fibrinogen-Albumin-Ratio Index (FARI), as currently reliable biomarkers to predict therapy response and prognosis of patients with advanced vulvar cancer are missing. Data of 124 consecutive patients, who underwent primary resection for vulvar cancer ≥pT1b, were retrospectively analyzed. Associations between FARI and disease recurrence were assessed fitting receiver operating characteristics (ROC) and binary logistic regression models; univariate and multivariable Cox regression models for disease-specific survival (DSS) and progression-free survival (PFS) were performed. A pretherapeutic low FARI cut at its median (<9.67) is significantly associated with younger age (65.5 vs. 74.0 years) and higher risk of recurrence (52.4% vs. 26.2%). The ROC analysis calculates the area under the curve (AUC) of the FARI for a PFS<6 months of 0.700 and for a DSS<12 months of 0.706, outperforming fibrinogen and albumin alone. The FARI remained independently predictive for PFS (HR 0.84, 95%CI [0.99-1.03], p=0.009) and DSS (HR 0.82, 95%CI [0.70-0.99], p=0.019) also in multivariable survival analysis. Despite the FARI’s promising predictive and prognostic value, however, further elucidation of its precise mode of action is warranted before clinical application as it appears to rely only on subtle changes of fibrinogen levels. However, there is concern which must be taken into account before the work can be reconsidered for publication. 

Can you please cite the references about FARI?  Lane 76: FARI was calculated as the ratio of fibrinogen (g/l) divided by albumin (g/l) multiplied by 100.

Author Response

Point 1: Can you please cite the references about FARI?  Lane 76: FARI was calculated as the ratio of fibrinogen (g/l) divided by albumin (g/l) multiplied by 100.

Response 1: We may thank the reviewer for this remark. We added reference no. 17 in line 82. Whereas the FARI has been evaluated in several solid malignant tumors to date (as highlighted in lines 49-51), Tan et al. was the first to report the FARI – as defined as the ratio of Fibrinogen (g/l) / Albumin (mg/dl) (different unit references as defined by the assays) to be independently prognostic for overall survival for esophageal squamous cell carcinoma back in 2017. As we homogenized fibrinogen and albumin units in g/l (table 1), we followed the approach of Chen at al., Clin Chim Acta 2018 (ref. 13) to depict the FARI as whole number for better readability and multiplied results for 100 eg. calculating with a FARI-cut off of </≥9.67 instead of </≥0.0967)

Reviewer 2 Report

Thanks for this interesting paper.

Author Response

We wish to thank the reviewer for the time and consideration of our manuscript.

Reviewer 3 Report

This retrospective study evaluated the role of Fibrinogen-Albumin-Ratio Index (FARI), in predicting response to therapy and prognosis in vulvar cancer patients. I have the following comments:

1.      The incidence rates of vulvar cancer in Austria should be added to the introduction section

2.      What are the most important prognostic factors in vulvar cancer?

3.      What guidelines have the researchers followed with regards to the treatment strategies? Are patients with locally advanced vulvar cancer always operated in the authors’ institution? If not, why did the authors not include patients who underwent primary chemoradiation?

4.      What percentage of cancers included in the study were associated with HPV infection and what percentage with lichen sclerosus?

5.      Please add a separate table (or add this information to Table 1) with an analysis of patients’ comorbidities. How do comorbidities influence this ratio?

6.      Page 8, line 227: How would the authors implement the findings of this study to clinical practice? Furthermore, how would the authors design a clinical study to further evaluate the role of this index?

Author Response

We wish to thank the reviewer for all the constructive comments, which provided valuable insight to refine the methodological details and reporting of results of our analysis

Point 1: The incidence rates of vulvar cancer in Austria should be added to the introduction section

Response 1: We may thank for this remark, which is particularly important, as vulvar cancer is likely to show different epidemiologic characteristics (considering an age-dependent incidence and an HPV-related etiology) depending on both the age distributions and cultural differences of different populations. We therefore added Austrian epidemiologic characteristics, which are comparable to aging populations of other industrial nations, to the introduction (line 33-35 reference no.4).

Point 2: What are the most important prognostic factors in vulvar cancer?

Response 2: Information on the most important prognostic factors of vulvar cancer (positive inguinal lymph nodes, FIGO stage, depth of invasion) was added to the introduction (incl. respective references 9 and 10, Zapardiel et al., VULCAN study IJGC 2020 and Woelber et al., Anticancer Res 2009) (lines 43-46)

Point 3: What guidelines have the researchers followed with regards to the treatment strategies? Are patients with locally advanced vulvar cancer always operated in the authors’ institution? If not, why did the authors not include patients who underwent primary chemoradiation?

Response 3: We thank the reviewer to raise this issue, which is particularly problematic for vulvar cancer, as solid evidence on optimal treatment strategies is limited due to the low incidence of the disease. Our department follows the ESGO guidelines. After primary staging with special attention to inguinal/pelvic lymph node metastases, we always aim to achieve radical local excision including surgical staging, if feasible. Patients considered unresectable by the tumorboard (e.g., extensive local tumorload, which would necessitate an exenteration while distant metastasis are present; infiltration of the pelvic wall) are referred to chemoradiotherapy.

In the present study, we tried to provide a cohort as homogenous and as close to clinical reality as possible to achieve maximal statistical power and to provide observations of clinical relevance. Therefore, decided on strict inclusion criteria and included all patients who were scheduled for primary resection and considered the final tumor stage in statistical analyses to avoid selection bias. Primary unresectable cases are relatively rare, even more neoadjuvant therapy approaches. If included, the cohort would have been heterogeneous; statistical power would not have sufficed for statistical analysis or comparison of such rare cases, necessitating sub-group analysis for patients undergoing primary surgery to avoid bias while not providing clinically meaningful results for the few patients with unresectable disease.

Point 4: What percentage of cancers included in the study were associated with HPV infection and what percentage with lichen sclerosus?

Response 4: We may thank the reviewer for this particularly valuable suggestion. We totally agree that the HPV status could be an important confounder on FARI values, as a different immunogenicity of HPV-dependent and non-HPV dependent tumors was previously proposed. However, as highlighted in the limitations section (lines 245-250), we could not provide comparative data on the tumors’ HPV-status due to the long observation period and changes in diagnostic standards within the timeframe (e.g., HPV-PCRs were unavailable for clinical routine in the early 2000s). Moreover, as our department is a tertiary care center treating patients from all over Austria for this particular scarce tumor entity, respective information (e.g., on previous treatment or diagnosis of lichen sclerosus in other centers or private practices in case advanced cancers were admitted) is not always available.

To address this clinically highly relevant question, we re-reviewed all histopathologic results for information regarding HPV-related alterations (e.g., description of koilocytes) or detection of lichen sclerosus. Respective data could be retrieved for 55 patients (44.4%). Thereof, 45.5% of tumors were identified as HPV-related, 54.5% as non-HPV related. Of the latter group, 17 patients (56.7%) had documented lichen sclerosus. No significant association between HPV-status at primary surgery and FARI levels could be established p=0.789. We adapted table 1 to provide respective information regarding the HPV-Status.

Point 5: Please add a separate table (or add this information to Table 1) with an analysis of  patients’ comorbidities. How do comorbidities influence this ratio?

Response 5: As comorbidity is an important confounder of survival in vulvar cancer patients, who are typically of advanced age, we decided to provide the ECOG performance status to allow for semi-quantitative comparison of data. As suggested by the reviewer, we provided an updated version of table 1 and changed the depiction from ECOG ≤2/>2 to ECOG 0, 1, 2, 3 after data review regarding ECOG statuses to provide more detailed information. We thereby corrected a typing error of the relational operators (and respective checksums) in the former version of the table, which meant to depict <2/≥2 (in line with all further calculations ECOG 0/1 vs 2/3). Due to the depiction of ECOG statuses as 0, 1, 2, 3, we changed the means comparison from chi-squared to ANOVA, resulting in a change of the p-value from 0.048 to 0.012.

Updated results are depicted in table 1, which highlight, that the patient cohort with higher FARI values, which was also significantly older, include a small percentage of patients with considerable comorbidities (e.g. two patients with ECOG 3, one with hemiplegia after an ICB years before and another one with pulmonary hypertension and cardiac dysfunction). However, it remains unclear whether higher FARI values are in causal relation to more comorbidities, or if the group with higher FARI values was only associated with more comorbidities as those patients are older (and the correlation between age and fibrinogen is well-established). Of note, first, considering the typically advanced age of vulvar cancer patients, we chose to calculate PFS and DSS, which should alleviate potential confounding of age and/or comorbidity as intercurrent death would be censored. Second, ECOG performance status was included in cox regressions and did not show significant associations with PFS and DSS. Concluding from these observations, we may assume that a potential confounding effect of age and/or comorbidity would be comparably small and that the prognostic – and therefore clinically meaningful –effect of FARI outperforms the respective effect of age and/or comorbidity on PFS and DSS. Contrarily, as the FARI appears to be resilient against age and/or comorbidity, the index may prove clinically valuable in older or morbid patients, which would be interesting to test in a prospective setting.

A respective statement was added to the limitations (lines 253-262).

To back our observations, we also calculated a log-rank/ Kaplan Meier applying the ECOG cut at 0 and 1 versus 2 and 3 as factor, demonstrating no association with PFS (p=0.356) and DSS (p=0.477). We may provide these figures as supplements if requested by the reviewer.

Point 6: Page 8, line 227: How would the authors implement the findings of this study to clinical practice? Furthermore, how would the authors design a clinical study to further evaluate the role of this index?

Response 6: As vulvar cancer patients are often of advanced age at the time of diagnosis and survival varies significantly depending on FIGO stage and lymph node invasion, defining a readily available, low-cost independent prognostic factor such as the FARI may provide valuable information for pretherapeutic patient triage. Of note, the observation of the FARI being seemingly resilient against age and comorbidity may prove particularly useful for this question to individualize clinical care for an orphan disease, for which only limited clinical data is available to date.

While we agree that present results are to be considered hypothesis-generating and require further validation, the low incidence of vulvar cancer renders prospective study designs difficult. Therefore, we would consider a multicenter retrospective validation of present results the most feasible first step; if promising, either designing a prospective observational study or co-assessing albumin and fibrinogen in ongoing vulvar cancer trials (eg. in studies such as the KEYNOTE-158) could be possible options.

Round 2

Reviewer 1 Report

The revised manuscript “The Prognostic Value of the Fibrinogen-Albumin-Ratio Index (FARI) in Patients with Advanced Vulvar Cancer” have adequately addressed my previous concerns and the paper is now acceptable for publication.

Reviewer 3 Report

All comments have been addressed and I believe the manuscript is now suitable for publication